# Infant formula composition and educational performance: a protocol to extend follow-up for a set of randomised controlled trials using linked administrative education records

Maximiliane Verfürden ![ORCID],[1] Katie Harron ![ORCID],[1] John Jerrim,[2] Mary Fewtrell,[1] Ruth Gilbert[1]

¹Great Ormond Street Institute of Child Health, University College London, London, UK
²Institute of Education, University College London, London, UK

**Correspondence to**
Maximiliane Verfürden;
m.verfuerden@ucl.ac.uk

## ABSTRACT

**Introduction** The effect of infant nutrition on long-term cognition is important for parents and policy makers. However, most clinical trials typically have short follow-up periods, when measures of cognition are poorly predictive of later function. The few trials with longer-term follow-up have high levels of attrition, which can lead to selection bias, and in turn to erroneous interpretation of long-term harms and benefits of infant nutrition. We address the need for unbiased, long-term follow-up, by linking measures of educational performance from administrative education records. Educational performance is a meaningful marker of cognitive function in children and it is strongly correlated with IQ. We aim to evaluate educational performance for children who, as infants, were part of a series of trials that randomised participants to either nutritionally modified infant formula or standard formula. Most trialists anticipated positive effects of these interventions on later cognitive function.

**Methods and analysis** Using data from 1923 participants of seven randomised infant formula trials linked to the English National Pupil Database (NPD), this study will provide new insights into the effect of nutrient intake in infancy on school achievement. Our primary outcome will be the mean differences in z-scores between intervention and control groups for a compulsory Mathematics exam sat at age 16. Secondary outcomes will be z-scores for a compulsory English exam at age 16 and z-scores for compulsory Mathematics and English exams at age 11. We will also evaluate intervention effects on the likelihood of receiving special educational needs (SEN) support. All analyses will be performed separately by trial.

**Ethics and dissemination** Research ethics approval, and approval from the Health Research Authority Confidentiality Advisory Group, has been obtained for this study. The results of this study will be disseminated to scientific, practitioner, and lay audiences, submitted for publication in peer-reviewed journals, and will contribute towards a PhD dissertation.

## Strengths and limitations of this study

► This study uses seven infant formula trials conducted in England to determine whether a range of nutritional interventions during infancy affects educational performance.

► We showcase the research potential of linking extant trials to administrative data, which can offer a low cost way to extend follow-up of early nutrition trials, maintain high rates of follow-up and safeguard confidentiality.

► There is no agreement on what defines educational performance and at what age it is best measured to determine intervention effects. We therefore rely on data from two objectively assessed, compulsory examinations sat by the majority of state school students in England.

cognitive function to inform infant feeding recommendations and guide parents. However, the majority of randomised controlled trials (RCTs) report only short-term effects of formula composition on proxy endpoints for cognition such as the Bayley Scales of Infant Development before 2 years of age.[1] Such early measures are crude, highly observer dependent and have poor predictive properties for later academic and employment outcomes.[1] Better measures such as IQ scores are available but limited by methodological issues such as small sample sizes, high attrition and no intention-to-treat analyses.[2 3]

Advances in the availability and quality of administrative datasets have created new possibilities for obtaining long-term outcomes on children's educational performance, which is a good predictor of future academic and employment opportunities.[4] Extending extant trial cohorts with educational performance data is achievable where trial data and

## INTRODUCTION
### Rationale

There is a need for high quality evidence on the long-term effects of infant formula on

BMJ

participant identifiers have been retained and governance arrangements allow secure linkage without the need for participants to re-engage. Educational performance is highly correlated with IQ scores[5] but socially more relevant than IQ scores as it determines future academic and employment opportunities.[6] Linkage to administrative records is available for a fraction of the cost compared with setting up new RCTs or using traditional follow-up methods and could offer a more rapid, complete and therefore less biased way to determine long-term intervention effects.[7 8]

We sought to determine long-term effects of modified infant formulas by linking RCTs to educational performance data for seven existing trial cohorts that compared modified infant formula interventions with standard infant formula.

### Research objective 1

To determine the effect of nutritionally modified infant formula on educational performance in standardised Mathematics and English language tests at ages 11 and 16.

### Research objective 2

To assess whether infant formula modifications have an effect on the risk of receiving special educational needs support during school.

### Research objective 3

To explore whether the effect of infant formula modifications on educational performance differs for boys and girls, by maternal smoking status during pregnancy, birth weight or maturity at birth.

## METHODS

### Data sources and linkage

We will follow 1923 participants from seven separate infant formula trials that randomised infants between 1993 and 2002 and were conducted in England (figure 1). Full details on the individual trial cohorts as well as previously reported outcomes can be found elsewhere (online supplementary table 1).

Outcome data for this follow-up will be retrieved from the National Pupil Database (NPD), an administrative data resource containing pupil-level and school-level data on all pupils in state schools in England and held by the Department for Education.[9] The Fischer Family Trust[10] (FFT) identified pupil records for the trial participants in 2018 on the basis of identifying information collected at randomisation and during periods of active follow-up (figure 2, online supplementary table 2).

### Interventions

The seven parallel group trials investigated the superiority of different infant formula modifications over standard formula in different infant populations (figure 3). The formula compositions for each trial are detailed in the online supplementary material. All but two of the

interventions (sn-2 Palmitate and Nucleotides) were assumed to have an effect on long-term cognitive function. All analyses will be stratified by trial as effect mechanisms might differ.

### Outcomes

#### Primary outcome

At age 16 years (during Key Stage 4), English pupils take their General Certificate of Secondary Education (GCSE). The primary outcome for this study will be the mean difference in GCSE mathematics exam z-score between control and intervention groups. GCSE mathematics exams are compulsory, nationally administered and are graded from 58 points to 0 point (table 1).

We chose Mathematics over English, which is also compulsory and nationally administered, because exam results for mathematics are commonly considered to be less subjectively graded.[11] We chose the primary endpoint at age 16 rather than age 11 as it is a more relevant predictor of future education and employment opportunities. Children who have not yet attended Key Stage 4 at the time of linkage will not be included in the primary outcome.

#### Secondary outcomes

As secondary outcomes, we will investigate intervention effects of modified infant formula versus control formula on:
► Mean GCSE English language exam z-scores.
► Mean Mathematics and English reading exam z-scores at age 11 (Key Stage 2, final year of primary school).
► Probability of receiving five or more GCSEs with grades A* to C, which includes Mathematics and English. (This is a critical accountability measure for schools and is used in school performance tables).
► Probability of receiving special educational needs (SEN) support.

All mentioned Mathematics and English exams are compulsory and nationally administered. GCSE English language scores are graded like GCSE Mathematics scores. Mathematics exams at age 11 are graded from 0 to 100, with 100 being highest possible score. English reading exams at age 11 are graded from 0 to 50, with 50 being highest. Receiving five or more GCSEs with grades A* to C is a commonly reported measure of academic performance as this measure feeds into entry requirements for a large number of sixth form colleges and therefore determines future academic development. Because the effect of the intervention on exam results might be mediated by special needs status, we also aim to determine whether the intervention affects the probability of receiving SEN support.

All z-scores will be calculated as:

$$Z - score = \frac{x - \mu}{SD_{pooled}}$$

where $SD_{pooled}$ is the pooled SD of the respective exam point scores from control and intervention groups within each trial.

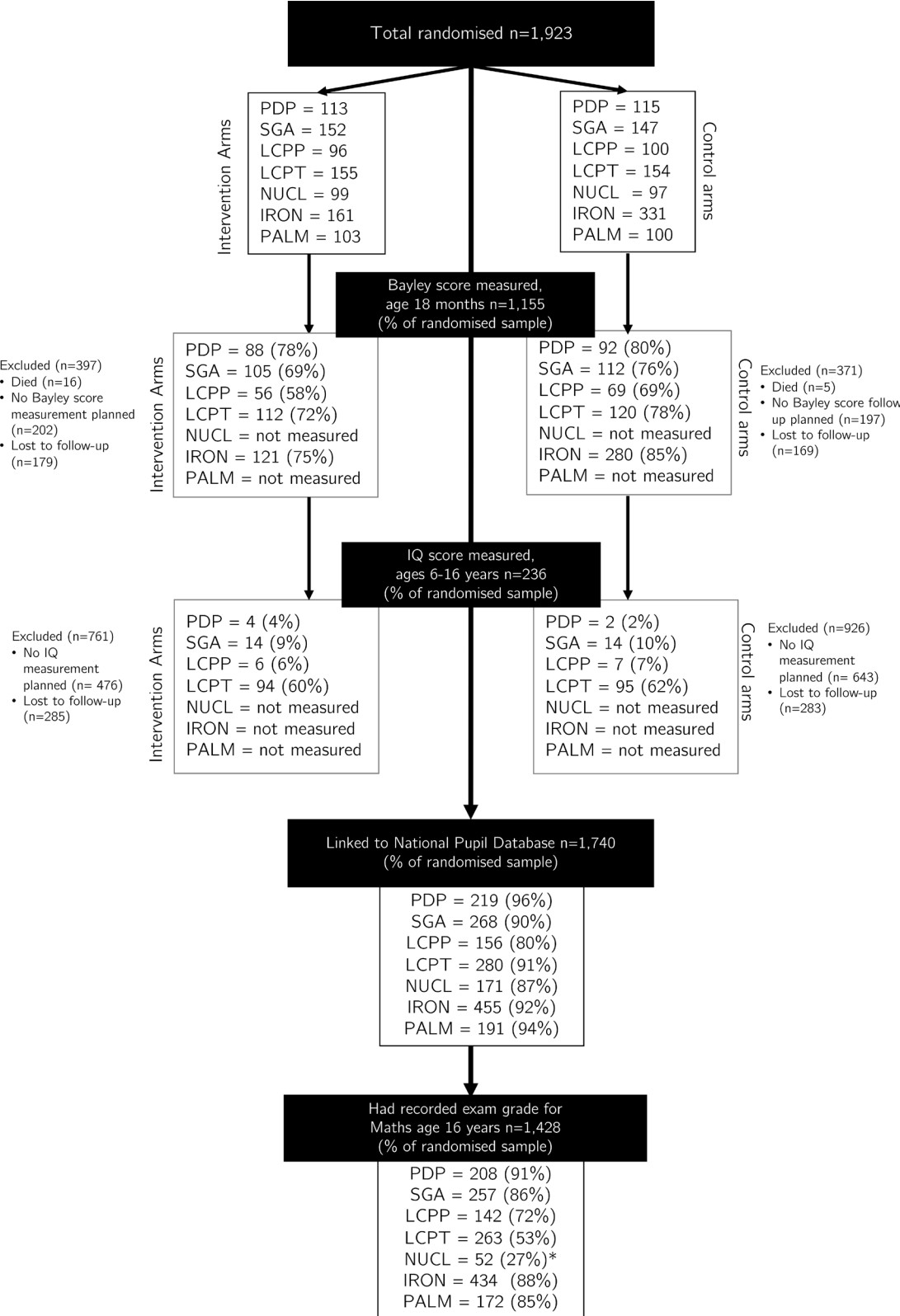

**Figure 1** Combined flow of participants of seven infant formula trials. Data analyst currently blinded to group allocation. PDP, Nutrient enriched formula post-hospital discharge for preterms, randomised 1993-96; SGA, Nutrient enriched formula for babies born small for gestational age, randomised 1993-96; LCPP, Long-chain polyunsaturated fatty acid enriched formula for preterms, randomised 1993-96; LCPT, Long-chain polyunsaturated fatty acid enriched formula for terms, randomised 1993-95; NUCL, Nucleotide enriched formula for terms, randomised 2000-2001; IRON, Iron enriched formula for terms, randomised 1993-94; PALM, Sn-2 Palmitate enriched formula for terms, randomised 1995-96. *Majority of RCT 5 participants is a year too young to have sat the Maths exam at age 16 at the time of data collection. RCT, randomised controlled trial.

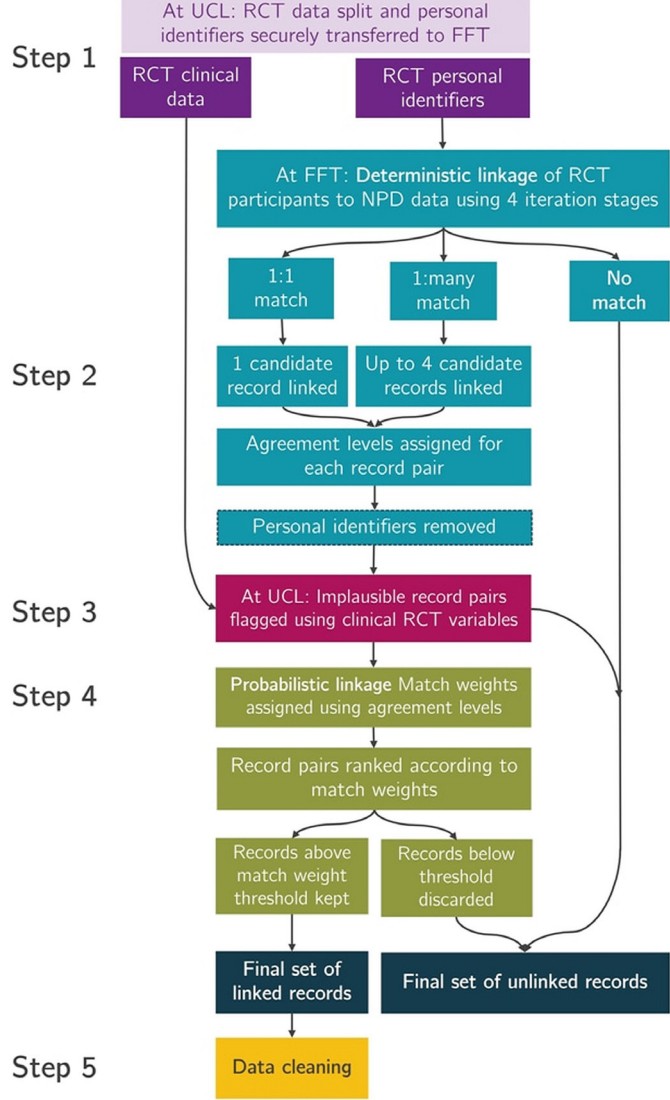

Step 1

Step 2

Step 3

Step 4

Step 5

**Figure 2** Data flows: (1) Participant identifier information securely transferred for linkage to FFT. (2) Deterministic linkage at FFT, application of agreement flags to all candidate pairs and candidate pairs securely sent to UCL minus participant identifiers. (3) Implausible pairs discarded using RCT clinical variables (linked through unique study ID number). (4) Probabilistic linkage at UCL. (5) Data cleaning. FFT, Fischer Family Trust; NPD, National Pupil Database; RCT, randomised controlled trial; UCL, University College London .

Probabilities will be reported as odds of having the outcome in the intervention group over odds of having the outcome in the control group within each trial.

### Covariates and interactions

To increase the statistical efficiency of our analysis, we will adjust for covariates that have a strong association with the analysis outcome and were measured at randomisation (ie, are not on the causal pathway between intervention and outcome). On the basis of previous internal analyses using linked Millennium Cohort Study and NPD data as well as previously published studies,[4 12 13] we expect

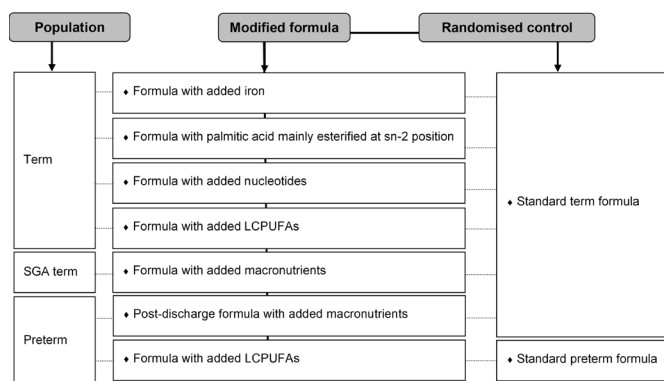

**Figure 3** Trial interventions, comparators and populations. LCPUFA, long-chain polyunsaturated fatty acids; SGA, born small for gestational age.

maternal smoking and maternal education as well as birth weight and gestational age to be strongly associated with educational performance. Analyses involving multicentre trials with separate randomisation schedules for each centre will also be adjusted for centre.[14] However, not every covariate was measured in all trials (table 2).

Based on reports from previous studies,[15] we also hypothesise that the effect of supplementations will depend on sex and whether the mother smoked during pregnancy or not. However, the trials were not powered to measure this interaction; therefore, we will include an interaction term in an exploratory analysis so the results can be included in future meta-analyses.

### Statistical methods

All statistical analyses will be conducted using Stata V.16.[16]

### Primary analysis

For our primary analysis, we will use multivariable regression models to compare intervention and control groups stratified by trial and adjusted for a priori determined risk factors (table 2). We will present effect estimates as mean differences with 95% CIs. The analysis will be on intention-to-treat basis. The primary analysis can be expressed using the following equation:

**Table 1** GCSE grade structure during analysis period (2001–2016)

| Grade | Points equivalent | Remarks |
|---|---|---|
| A* | 58 | Highest pass |
| A | 52 | |
| B | 46 | |
| C | 40 | |
| D | 34 | |
| E | 28 | |
| F | 22 | |
| G | 16 | Lowest pass |
| U | 0 | Ungraded |

GCSE, General Certificate of Secondary Education.

**Table 2** Variables a priori expected to be associated with educational performance

| Trial | Centre | Gestational age | Birth weight | Maternal smoking | Maternal education |
|---|---|---|---|---|---|
| Type | Nominal | Continuous | Continuous | Binary | Ordinal |
| RCT 1 1993/96 Nutrient-enriched post-dc | X | X | X | X | X |
| RCT 2 1993/96 Nutrient-enriched (SGA) | X | X | X | X | X |
| RCT 3 1993/96 LCPUFA (preterm) | X | X | X | X | X |
| RCT 4 1993/95 LCPUFA (term) | X | X | X | X | X |
| RCT 5 2000/02 Nucleotides | X | X | X | X | X |
| RCT 6 1993/94 Iron | X | X | X | X | X |
| RCT 7 1995/96 Sn-2 palmitate | n/a | X | X | X | X |

LCPUFA, long-chain polyunsaturated fatty acids; n/a, not applicable; post-dc, formula after discharge from hospital; RCT, randomised controlled trial; SGA, born small for gestational age.

$$Z-\text{score}_{GCSEMaths} = \beta_0 + \beta_1 \times centre + \beta_2 \times sex + \beta_3 \times birthweight +$$
$$... + \beta_p \times controlmilk + e$$

where $\beta_0$ (intercept) is the z-score of GCSE Mathematics when all predictors are zero and $e$ refers to the residual terms. This model assumes that GCSE Mathematics scores are linearly related to the other predictors. If for a given predictor the relationship is not linear, we will account for this by categorising quantitative variables or including quadratic terms. We will also check that there is no multicollinearity between the predictors, that the residuals are consistent with random error and independent, and that the variance is constant across the outcome and the predictors.

### Secondary analyses

Mean GCSE English language exam z-scores and mean Mathematics and English reading exam z-scores at age 11 will be analysed in the same way as the primary outcome.

The probability of receiving five or more GCSEs with grades A* to C and the probability of receiving SEN support will be modelled using logistic regression and adjusted for the same factors as the primary outcome (table 2). We will take into account the risk of false positives arising from multiple testing when interpreting the results. Line graphs will be used to visually explore Mathematics and English exam z-scores and their 95% CIs at ages 16 and 11 by trial and trial arm.

### Sensitivity analyses

To determine the effect of the intervention on GCSE Mathematics scores in context of the general population, we will calculate the z-scores using national SD rather than the trial-specific ones. Due to the higher heterogeneity in the national sample, we expect our effect estimate to be more conservative in this analysis.

In order to assess the robustness of our analysis, we will also present the unadjusted outcomes as a sensitivity analysis and model the outcome using an ordered logit model adjusting for the above mentioned covariates (table 2).

To assess the sensitivity of our endpoint measure, we are going to present results for the cohort of non-randomised breastfed recruits from the trials alongside the randomised groups. Breastfed children have been consistently reported to have better cognitive ability and perform better in school than their formula-fed peers.[17 18] Therefore, we also expect to observe better school performance for previously breastfed compared with formula-fed children in the primary outcome. Breastfed children were recruited in all trials except for the 1993/94 iron trial. Additionally, we expect there to be no effect of sn-2 palmitate-enriched formula and included this trial cohort to further validate our approach.

### Exploratory analyses

While this study is underpowered to determine interaction effects, we intend to conduct exploratory analyses stratified by sex and smoking status during pregnancy, so these are available for inclusion in future meta-analyses.

### Power

The primary outcomes (mean difference in z-score of Mathematics exam at age 16) will be continuous and analyses will be conducted for each trial separately. We expect primary outcomes to be available for around 70% of the original cohort with an equal retention rate in treatment and control arms. A 0.1 SD difference (this corresponds to one-fifth of a GCSE grade or 1.154 points on the 58 points point scale) will be regarded as clinically significant.[19 20] The desired power to detect a difference between the treatment arms that is at least as large as this

is 80% with a significance level of 0.05. The R-squared is expected to be 0.05–0.1. Taking the 1993/95 LCPUFA term study (n=309) as an example, a conservative 50% (n=154) linkage success would enable a detection of 0.45 SD difference (eg, 9/10 of a grade or 5.193 grade points).

## Missing data

For our primary analysis, we will use multiple imputation to impute missing covariates using 15 imputations and setting the random seed at 300.[14] We will use Stata's *mi chained* command to perform the imputation. We will also perform sensitivity analyses to determine the potential impact of missing outcome data by looking at worst case scenarios (ie, all missing participants with missing outcome data achieved the lowest grade/highest grade possible).

## Assessment of bias

For each trial, we will compile information for a Cochrane risk of bias analysis.[21]

To detect whether selection bias at follow-up has affected the randomisation balance of characteristics, we will present the prerandomisation characteristics as percentage/mean differences between intervention and control groups at randomisation and at age 16 (when the primary outcome is measured) with 95% CIs.

## DISCUSSION
### Expected outcomes of the study

High participant retention rates are essential to determine delayed intervention effects that might not become apparent in short-term follow-ups or when using traditional follow-up methods, which typically suffer from high attrition.[2 22] Extending follow-up of RCTs without the need for participants to re-engage is possible where trial data and participant identifiers have been retained and governance arrangements allow secure linkage that safeguards participants' privacy. Post-trial extension using opt-out linkage to administrative data will be especially desirable for future research if it reduces costs associated with conventional methods of follow-up (such as face-to-face interviews) and simultaneously improves participant retention rates (compared with opt-in data linkage). Critically, our study will allow new insights into the long-term cognitive benefits and safety profile of modified infant formulas, by making use of randomised historical infant formula comparisons, which are unlikely to be investigated in future RCTs either due to lack of equipoise or resources.

## Limitations

Our study should be considered in the context of several limitations.

First, only schools following the national curriculum have to contribute data to the NPD, while independent schools may provide data voluntarily.[23] Some participants who received their education outside state-funded English schools will therefore be excluded from this analysis. In

any given academic year, however, data from 99% of all children of compulsory school age are included in the NPD.[9] Consequently, we expect this to have minimal, non-differential impact on overall linkage success.

Second, there is no general agreement on what defines educational performance and which age is optimal to capture intervention effects. To minimise missing data and detect potential trajectory effects, we will therefore rely on data from two high-stakes examinations sat by the majority of state school students in England.

Third, linkage quality plays a role when using administrative data to follow up participants.[24–27] The FFT will provide us with up to three different pupil records per participants (based on names, date of birth and location). We will use probabilistic methods adapted from Fellegi and Sunter[28] as well as manual review to decide which record, if any, we consider to be the best match (online supplementary material).

Fourth, we acknowledge that intervention effects measured 15 years later might be diluted by contextual factors. Convergence of trial arms may result from catch-up over time as those who lag behind in early years might receive extra (eg, educational) assistance. There will also be variability in terms of quality of education. However, this is likely to be roughly equal between intervention and control groups due to adequate sample size, randomisation and blinding. Furthermore, theories of early programming suggest that early postnatal life is a critical period of development[29] and nutritional stimuli during this period can result in lifelong permanent effects.[30] If true, any effective nutritional intervention is unlikely to disappear completely by early adulthood. If not, our study should be of increased interest to the scientific community concerned with the early origins of adult health and disease theory. Either way, it will provide important insights for parents, policy makers and the scientific community as to whether choice of infant formula type can have long-lasting, measurable effects on something as important as educational performance.

## ETHICS AND DISSEMINATION
### Ethics approval and legal basis for data processing

The proposal to link the trials and NHS data underwent a robust ethical, legal and technical review. The proposed project was approved by the UCL Research and Development Office and Data Protection Office (Reference: 14PE15), the NHS London-City and East Research Ethics Committee (Reference: 17/LO/0556) as well as the Health Research Authority and Health and Care Research Wales (Reference: 212148). An exemption under Section 251 of the NHS Act 2006 was granted by the Confidentiality Advisory Group (Reference: 17CAG0051) and allows UCL GOS ICH and the FFT to process identifiable participant information to facilitate the data linkage and analyses without consent. Dissent and fair processing were discussed with the HRA CAG and an opt-out procedure is in place.[31]

## Confidentiality

Special efforts are made to protect participant privacy and confidentiality. All linked data will be de-identified and thus contain no information on personal identifiers such as names and addresses. Identifiers will be stored at separate access-restricted locations certified to international standards (ISO/IEC 27001). Access to all data will also be physically restricted. Analysis data will be stored as aggregated as possible (such as school types instead of school IDs), to minimise the risk of re-identification. All analyses will be conducted in a digital safe haven and no individual-level data is allowed to leave the server. Outputs will have to meet statistical disclosure controls that prevent small cell sizes to be exported from the server. Additionally, access to the data will be restricted to an authorised group of researchers, who will all have undergone the HSCIC and ADRN information governance training and are bound by their university contracts to treat the data confidentially.

## Project management

The Fischer Family Trust links the data, which is then securely transferred to the UCL data safe haven. MV will clean and analyse the linked data and draft the initial manuscript. KH and RG will advise on data cleaning and contribute to the interpretation of results. JJ will contribute to interpreting exam results and MF will provide background information on the trials and contribute to interpreting potential effects of the modified formula. All authors will contribute to writing the final manuscript.

## Dissemination of results and publication policy

To enhance the use, understanding and dissemination of this study, the results will be disseminated to both scientific and lay audiences, submitted for publication in peer-reviewed journals, and will contribute towards a PhD dissertation.

**Acknowledgements** The authors thank the participants of the trials and their parents as well as the original investigators for the valuable time they have invested in generating this data. We also thank the peer reviewers for their valuable comments, which in our view havesignificantly improved the quality of our protocol.

**Contributors** MV has prepared the initial draft of the protocol. KH, JJ, MF and RG provided substantial intellectual input. All authors have contributed to writing the final draft of the manuscript.

**Funding** MV is supported by the Economic and Social Research Council (ESRC) and the Great Ormond Street Hospital Charity. RG receives NIHR funding as senior investigator and through the NIHR policy research programme. RG and KH receive funding from Health Data Research UK, an initiative funded by the UK Research and Innovation, Department of Health and Social Care (England) and the devolved administrations, and leading medical research charities. KH receives funding from Wellcome (grant 103975/Z/14/Z).

**Competing interests** MF has been a member of the Infant Nutrition Working Group at EFSA (European Food Safety Authority) since 2013. She was involved in data analysis and publication of randomised trials of LCPUFA-supplemented infant formulas funded by grants from Numico Res BV and Heinz UK. The companies also provided the infant formulas for the studies. She was also involved in follow-up studies (including cognitive outcome) of children and adolescents from randomised trials of LCPUFA-supplemented formulas, with funding from the Medical Research Council and European Union (FP6-FOOD-2005-0 07 036).

**Patient and public involvement** Patients and/or the public were involved in the design, or conduct, or reporting, or dissemination plans of this research. Refer to the Methods section for further details.

**Patient consent for publication** Not required.

**Provenance and peer review** Not commissioned; externally peer reviewed.

**ORCID iDs**
Maximiliane Verfürden http://orcid.org/0000-0003-2204-8251
Katie Harron http://orcid.org/0000-0002-3418-2856

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
