## [Reviewer comments · BMJ Open]

ARTICLE DETAILS

TITLE (PROVISIONAL)	Infant formula composition and educational performance: a protocol to extend follow-up for a set of randomised controlled trials using linked administrative education records
AUTHORS	Verfürden, Maximiliane; Harron, Katie; Jerrim, John; Fewtrell, Mary; Gilbert, Ruth

VERSION 1 – REVIEW

REVIEWER	Jeffrey Roth University of Florida USA
REVIEW RETURNED	12-Dec-2019

GENERAL COMMENTS	This protocol proposes to compare test scores on a compulsory Math exam at age 16 to an estimated 2157 pupils at English schools who as infants participated in one of seven randomized controlled trials of infant formula supplementation. The purpose of this linkage study is to provide long term follow-up information about the cognitive performance of the children who participated in these trials between 1993 and 2001. The study will also compare scores of these infant formula trial participants on a compulsory English test at 16, Math and English compulsory test scores at 11, and the incidence of receiving special education services. Descriptions of the proposed data cleaning methods, matching records estimation, statistical and sensitivity analyses are clear and sound. The authors are correct that this study, when concluded, may be able to offer intriguing information about the potential efficacy of various modifications to standard infant formula. The main concern I have with this longitudinal study is that it does not consider important information about the cumulative academic experiences of these subjects: 1) the kinds of schools they attended (e.g., proportion of students in poverty, average years of teachers' experience; district per pupil funding); 2) whether they participated in preschool, early intervention, remedial, or gifted programs; 3) secular changes to curriculum and/or compulsory exam content that occurred over the course of subjects' 10 years' enrollment in school. Absent some accounting for these contextual factors, any association posited between nutrition in the first year of life and performance on a math test fifteen years later will perforce have to be highly tenuous. That said, many parents and educators may be interested to know whether infants whose formula was fortified with a protein, fat, carbohydrate, or mineral perform differently when tested at mid-adolescence. Given a possible interest in this stretched-out associational thread between early nutrition and
---

	later cognitive ability, I suggest some clarifications for world-wide readers of BMJ Open who may be unfamiliar with specifics of state schooling in England:  • Are the “five or more GCSEs with grades A* to C” [P7, L3] in subjects other than Math or English language? If so, please tell us what those are. Is Grade A* equivalent to A+? • References are needed to back up statements such as “educational performance is . . . strongly correlated with intelligent quotient” [P3, L18-20]; “early measures . . . have poor predictive properties for later academic and employment outcomes” [P5, L15-17]; “the sn-2 Palmitate intervention cohort was included . . . with an expected null-effect on educational performance” [P6, L 46-48] • Be sure to specify the substances included in “macro nutrients and vitamins” {Supplementary Table 1} • Can you tell us what might be the educational impact (“clinically significant,” P10, L36) of a 1 point difference on a 58 points point scale?
--	--

REVIEWER	John Robert Bautista The University of Texas at Austin
REVIEW RETURNED	16-Apr-2020

GENERAL COMMENTS	Please see the following comments/suggestions:  1. The consent for infant trials are with their parents/guardian. Based on the write-up, it is mentioned that data linkage is seen as a method to "allow secure linkage without the need for participants to re-consent." However, the issue is that the infants were included in the trial because their parent/guardian consented to it. For obtaining scores, should we consider consent from the parent/guardian and with the offspring (who is now an adolescent)? 2. Maybe add RO1 for research objective 1 (primary) and RO2 and RO3 for the secondary objectives. This will help you create subsections when reporting results in the future. 3. Is the third objective part of the study to be carried out or you are just pointing out that it will be used in the future. May need to delete this one if this is only part of the discussion section of the completed study. Your manuscript does not also refer to any specific analysis for the 3rd objective so might as well remove that. 4. Maybe mention the software to be used for the analyses. Only Stata was mentioned in the missing data section. 5. You need to provide some citations on the points you mentioned in the discussion section.
--

REVIEWER	Josie Athens University of Otago, New Zealand
REVIEW RETURNED	22-Apr-2020

GENERAL COMMENTS	Very well presented protocol. The aims of the study are clear, outcomes well detailed and justified, limitations addressed and the methodology is appropriate. My only feedback is for the authors to consider a post-hoc analysis to adjust confidence intervals for multiple comparisons. Post-hoc analysis was not addressed in methods.
---

VERSION 1 – AUTHOR RESPONSE

Response to reviewer 1:

Nr	Comment	Response	Location of change
1.1	The main concern I have with this longitudinal study is that it does not consider important information about the cumulative academic experiences of these subjects: 1) the kinds of schools they attended (e.g., proportion of students in poverty, average years of teachers' experience; district per pupil funding); 2) whether they participated in preschool, early intervention, remedial, or gifted programs; 3) secular changes to curriculum and/or compulsory exam content that occurred over the course of subjects' 10 years' enrollment in school. Absent some accounting for these contextual factors, any association posited between nutrition in the first year of life and performance on a math test fifteen years later will	perform differently when tested at mid-adolescence. performance have to be highly tenuous. That said, many parents and educators may be interested to know whether infants whose formula was fortified with a protein, fat, carbohydrate, or mineral We agree with the reviewer that p.13, lines 4-16 intervention effects can fade out over (Discussion) time. Convergence of trial arms may result from catch-up over time as those who lag behind in early years might receive extra (e.g. educational) assistance. There will also be variability in terms of quality of education. However, this is likely to be roughly equal between intervention and control groups due to adequate sample size, randomisation, and blinding. Furthermore, theories of early programming suggest that early postnatal life is a critical period of development¹ and nutritional stimuli during this period can result in lifelong permanent effects². If true, any effective nutritional intervention is unlikely to disappear completely by early adulthood. If not, this study will be of increased interest to the scientific community concerned with the early origins of adult health and disease theory. We therefore agree with the reviewer, that either way, parents, policy makers and the scientific community will be eager to learn whether choice of infant formula type can have long-lasting, measurable effects on something as important as educational performance.	
	perform differently when tested at mid-adolescence. discussion	We do thank the reviewer for raising this point and have added a discussion to the protocol.	
1.2	Are the "five or more GCSEs A* to C" [P7, L3] other than Math or English language? If so, and please tell us what those are. Is Grade A* equivalent to achievement A+?	We thank the reviewer for this comment and have now clarified the grade structure, adding an extra table translating the grades into points percentages, and an explanation of the choice of measuring of 5+ GCSEs.	p.7, table 1 with grades lines 20-22 in subjects (Methods)
1.3	References are needed to back up statements such as "educational performance is	Thank you, we have added references for all statements except for the palmitate statement. No studies, . .	

. *strongly correlated with* observational or experimental, have *intelligent quotient* [P3, L18- been conducted that test the 20]; *“early measures . . . have association between sn-2 palmitate poor predictive properties for and markers of cognition and to date later academic and there has been no proposition that this employment outcomes”* [P5, association is biologically plausible. L15-17]; *“the sn-2 Palmitate intervention cohort was included . . . with an expected null-effect on educational performance”*

[P6, L 46-48]

1.4 *Be sure to specify the substances included in “macro nutrients and vitamins” {Supplementary Table 1}*

We thank the reviewer for highlighting this omission. We have now added all trial formula compositions to the supplementary material.

material
p.5,
line 8

p.5, line 19
(Rationale)

Supplementary

1.5 *Can you tell us what might be the educational impact* We thank the reviewer for the *the opportunity to discuss our choice of*

“clinically significant,” P10, the effect size we consider clinically L36) of a 1 point difference significant. There are several aspects on a 58 points point scale? to our decision: (1) In absence of bias,

the effect size of the presented

interventions would be causal and should not be compared to correlational effect sizes which tend to be inflated. (2) As with any early childhood intervention it is likely that effect estimates are subject to the fadeout effect. Re-emergence of early childhood intervention effects on attainment have been reported; however, they were smaller in size.³ In this light, we consider even a

comparatively small effect to be of clinical significance. (3) Critically, the

intervention (change of infant formula composition) is scalable at low cost and can potentially affect a large proportion of children in education. Raising grades of formula fed children by an average 1/5th of a grade than is currently standard for formula fed children would have a large population impact (4) Lastly, it is comparable to effect sizes reported by Fryer who found average effect sizes of 0.05 SD in math and 0.07 SD in reading based on 105 school-based RCTs.⁴

Response to reviewer 2:

Nr	Comment	Response	Location of change
2.1	The consent for infant trials are with their parents/guardian. Based on the write-up, it is mentioned that data linkage is seen as a method to "allow secure linkage without the need for participants to re-consent." However, the issue is that the infants were included in the trial because their parent/guardian consented to it. For obtaining scores, should we consider consent from the parent/guardian and with the offspring (who is now an adolescent)?	2.2 Maybe add RO1 for research We thank the reviewer for highlighting p.5, line 18 this inconsistency. We have now reworded this section. To clarify, we have received ethical approval to conduct this study not on the legal basis of consent but on the legal basis of section 251 of the English National Health Services act 2006. Our project was approved by the NHS London-City & East Research Ethics Committee (REC) as well as the Health Research Authority (HRA) and Health and Care Research Wales (HCRW). An exemption under Section 251 of the NHS Act 2006 was granted by the Confidentiality Advisory Group (CAG), a committee set up to advise the HRA on applications involving the sharing and processing of confidential patient data in cases where participant consent is not practicable to obtain. We discussed the issue of	(Rationale)

dissent and fair processing with the HRA CAG and have an opt-out procedure in place.

analyses by other researchers.

Thank you, we have now added **p.5-6**

objective 1 (primary) and RO2 subsections accordingly. (Rationale) : the secondary objectives. This will help you create subsections when reporting the future.

2.3 *Is the third objective part of the study to be carried out or you are just pointing out that it will be used in the future. May need to delete this one if this is only part of the discussion section of the completed study. Your manuscript does not also refer to any specific analysis for the 3rd objective so might as well remove that.*

We thank the reviewer for pointing this out. The third objective refers to the test for interaction that is described in more detail in the sections “covariates and interactions” and “exploratory analyses”. The trials we link were not powered for measuring an interaction but we are aware of a general scientific interest in these interactions. While we do not expect the interaction analyses to yield statistically significant results, reporting them in the protocol will allow the results to be identified and included in future meta-

2.4 *Maybe mention the software to used for the analyses. Only missing data section.* Thank you, we have added the software. **p.9, line 15 be (Statistical Stata was mentioned in the methods)**

2.5 *You need to provide some the points you mentioned in the discussion points.* We thank the reviewer for highlighting this and have now included more **(Discussion) citations on citations to support our discussion section.**

Response to reviewer 3:

Nr	Comment	Response	Location of change
3.1	My only feedback is for the authors to consider a post-hoc analysis to adjust confidence intervals for multiple comparisons. Post-hoc analysis was not addressed in methods.	We thank the reviewer for suggesting to address multiplicity. We have added this to our methods.	p.10, lines 6-7 (Statistical methods)

References:

1. Barker DJ, Osmond C. Infant mortality, childhood nutrition, and ischaemic heart disease in England and Wales. *The Lancet* 1986;327(8489):1077-81.
2. Lucas A. Long-term programming effects of early nutrition -- implications for the preterm infant. *J Perinatol* 2005;25 Suppl 2:S2-6. doi: 10.1038/sj.jp.7211308 [published Online First: 2005/04/30]
3. Bailey D, Duncan GJ, Odgers CL, et al. Persistence and Fadeout in the Impacts of Child and Adolescent Interventions. *J Res Educ Eff* 2017;10(1):7-39. doi: 10.1080/19345747.2016.1232459 [published Online First: 2016/11/14]
4. Fryer Jr RG. The production of human capital in developed countries: Evidence from 196 randomized field experiments. *Handbook of economic field experiments*: Elsevier 2017:95-322.

VERSION 2 – REVIEW

REVIEWER	Jeffrey Roth University of Florida U.S.A.
REVIEW RETURNED	06-May-2020
GENERAL COMMENTS	The authors have diligently amplified their manuscript in response to reviewer suggestions. New tables in the text, and in Supplemental Material add a high degree of specificity to what was

	already a carefully constructed protocol. The enlargement of the sections, Limitations, Ethics Approval, and Confidentiality, as well as the SPIRIT checklist, give readers every confidence that this study will be conducted with scientific precision. On rereading, I came across a few small matters which, when addressed, may reduce some readers' subsequent questioning.  • Consider changing “residuals are normally distributed” to “residuals are randomly distributed” [P47, L47-8] • The link in Reference 30 to the Head or Heart Study [https://www.ucl.ac.uk/child-health/research/population-policy-and-practice/research/studies/head-or-heart-study] states: “follow-up rates were low in later years, and therefore many important long-term outcomes could not be investigated.” Please provide some information about attrition rates in the original study cohorts, even though you are going to perform an assessment of selection bias [P49, L 31-8] • Does the statement that infant formula comparisons are “unlikely to be investigated in future RCTs . . . due to lack of equipoise” [P50, L16-9] mean it is now established which infant formula composition has been found to be superior? • What statistical method(s) will be employed to “detect potential trajectory effects” [P50, L40-42] • In last row of Supplemental Table 1, spell out acronyms DHA and AA • The Data Cleaning Methods describe how the 1,923 participants in the original seven randomized infant formula trials will be matched to the English National Pupil Database. Have you performed any pilot work that gives you confidence that the matching algorithms will yield a sufficiently large sample size, or, as stated in the protocol “maintain high rates of follow-up” [P41, L17]? • In Trial Formula composition tables for RCTs, specify difference from standard formula in column heading; e.g., in RCT 3, “LCPUFA-enriched” instead of “modified”
--	---

VERSION 2 – AUTHOR RESPONSE

Response to reviewer 1:

Nr	Comment	Response	Location of change
1.1	Consider changing “residuals are normally distributed” to “residuals are randomly distributed” [P47, L47-8]	Many thanks for identifying this. This has been amended.	p.9, Statistical methods

1.2	The link in Reference 30 to the Head or Heart Study [https://www.ucl.ac.uk/childhealth/research/populationpolicy-and-practice/research/studies/heador-heart-study] states: “follow-up rates were low in later years, and therefore many important long-term outcomes could not be investigated.” Please provide some information about attrition rates in the original study cohorts, even though you are going to perform an assessment of selection bias [P49, L 31-8]	We thank the reviewer for highlighting this omission. We have now added information on attrition rates to Supplementary Table 1.	Supplementary Table 1
1.3	Does the statement that infant formula comparisons are “unlikely to be investigated in future RCTs . . . due to lack of equipoise” [P50, L16-9] mean it is now established which infant formula composition has been found to be superior?	Thank you for flagging up the issue around equipoise. With this statement we refer in particular to the decision by the European Commission to mandate the addition of one type of LCPUFA, DHA, to all infant and follow-on formulae by 2020.¹ This decision acknowledged the lack of RCT evidence on cognitive benefits, but cited previously issued advice on nutrient requirements based on theoretical arguments and the absence of	No changes in manuscript
		harmful effects in previously published data to explain the mandate. So while superiority has not been established it is likely that in practice there is a perceived lack of equipoise which –coupled with high financial costs– makes it difficult to conduct new trials in this area.	
1.4	What statistical method(s) will be employed to “detect potential trajectory effects” [P50, L40-42]	We will use line graphs to visually explore Mathematics and English exam z-scores and their 95% confidence intervals at age 16 and 11 by trial and trial arm. This clarification has now been added to the methods.	p.9, Secondary analyses
1.5	In last row of Supplemental Table 1, spell out acronyms DHA and AA	Many thanks for identifying this. This has now been amended.	Supplementary Table 1

1.6	The Data Cleaning Methods describe how the 1,923 participants in the original seven randomized infant formula trials will be matched to the English National Pupil Database. Have you performed any pilot work that gives you confidence that the matching algorithms will yield a sufficiently large sample size, or, as stated in the protocol “maintain high rates of followup” [P41, L17]?	We based our estimates of linkage success on previous linkages: e.g. 82% accurate linkage has been reported for the ALSPAC cohort with English education data. ² To reflect the uncertainty about the results of our particular linkage we have now modified the article summary to “we showcase the research potential of linking extant trials to administrative data, which can offer a low cost way to extend follow-up of early nutrition trials, maintain high rates of follow-up, and safeguard confidentiality.”	Article summary Added reference to introduction
1.7	In Trial Formula composition tables for RCTs, specify difference from standard formula in column heading; e.g., in RCT 3, “LCPUFAenriched” instead of “modified”	We have clarified the headings according to the reviewer’s suggestion.	Supplementary material (trial formula compositions)

1. EU Commission. Commission Delegated Regulation (EU) 2016/127 of 25 September 2015 supplementing Regulation (EU) No 609/2013 of the European Parliament and of the Council as regards the specific compositional and information requirements for infant formula and follow-on formula and as regards requirements on information relating to infant and young child feeding. *OJEC* 2016;59:1-29.
2. Boyd A, Golding J, Macleod J, et al. Cohort Profile: The ‘Children of the 90s’— the index offspring of the Avon Longitudinal Study of Parents and Children. *Int J Epidemiol* 2012 doi: 10.1093/ije/dys064